# Investigating the Potential Impact of Future Climate Change on UK Supermarket Building Performance

Agha Hasan [1] , Ali Bahadori-Jahromi [1,*] , Anastasia Mylona [2], Marco Ferri [3] and Hooman Tahayori [4]

1    Department of Civil Engineering and Built Environment, School of Computing and Engineering, University of West London, London W5 5RF, UK; 21445082@student.uwl.ac.uk
2    Research Department, The Chartered Institution of Building Services Engineers (CIBSE), London SW12 9BS, UK; amylona@cibse.org
3    Lidl Great Britain Ltd., 19 Worple Road, London SW19 4JS, UK; marco.ferri@lidl.co.uk
4    Department of Computer Science & Engineering and IT, School of Electrical and Computer Engineering, Shiraz University, Shiraz, Iran; tahayori@shirazu.ac.ir
*    Correspondence: ali.bahadori-jahromi@uwl.ac.uk

**Abstract:** The large-scale shifts in weather patterns and an unprecedented change in climate have given rise to the interest in how climate change will affect the carbon emissions of supermarkets. This study investigates the implications of future climatic conditions on the operation of supermarkets in the UK. The investigation was conducted by performing a series of energy modelling simulations on a LIDL supermarket model in London, based on the UK Climate Projections (UKCP09) future weather years provided by the Chartered Institution of Building Services Engineers (CIBSE). Computational fluid dynamic (CFD) simulations were used to perform the experiment, and the baseline model was validated against the actual data. This investigation ascertains and quantifies the annual energy consumption, carbon emissions, and cooling and heating demand of the supermarket under different climatic projections, which further validate the scientific theory of annual temperature rise as a result of long-term climatic variation. The maximum percentage increase for the annual energy consumption for current and future weather data sets observed was 7.01 and 6.45 for the 2050s medium emissions scenario, (90th) percentile and high emissions scenario, (90th) percentile, respectively, and 11.05, 14.07, and 17.68 for the 2080s low emissions scenario, (90th) percentile, medium (90th) percentile and high emissions scenario (90th) percentile, respectively. A similar inclining trend in the case of annual $CO_2$ emissions was observed where the peak increase percentage was 6.80 and 6.24 for the 2050s medium emissions scenario, (90th) percentile and high (90th) percentile, respectively and 10.84, 13.84, and 17.45 for the 2080s low emissions scenario, (90th) percentile, medium emissions scenario (90th) percentile and high emissions scenario (90th) percentile, respectively. The study also analyses the future heating and cooling demands of the three warmest months and three coldest months of the year, respectively, to determine future variance in their relative values.

**Keywords:** energy performance; future weather; sustainability; building simulation

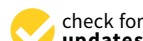

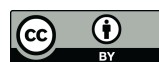

## 1. Introduction

The impact of $CO_2$ emissions within the built environment is a cause for alarm in the UK and globally, especially given the fact that there is little time to make a positive impact. The growing concern over increasing pollution and greenhouse gas (GHG) emissions has initiated a debate among the UK Government and other sectors of industry to reduce its environmental impact to ensure a sustainable future. As a result, the UK Government announced the Climate Change Act 2008, introducing a legally binding framework to cut emissions of greenhouse gasses by 80% by 2050 compared with 1990 levels as defined in chapter 27 of the act [1]. The increase in global $CO_2$ and GHG emission levels is mainly attributed to anthropogenic activities as an industrial production system and economic

development relies mainly on burning fossil fuels, damaging the atmosphere and the carbon cycle [2,3].

There is abundant evidence to suggest that global warming is the main contributor to the increase in climatic change [4] and it has an adverse effect on the built environment as it will directly affect the cooling and heating demand of the buildings. The most recent decade (2010–2019) has been on average 0.9 °C warmer across the UK than the period 1961–1990, with 2019 being 1.1 °C above the 1961–1990 long-term average [5]. As for 2019, it was the sixth consecutive year with fewer frosts than average, and it was one of the least snowy years on record. The year 2019 was most remarkable for setting four UK high-temperature records [5], including the following:

- A new record (38.7 °C), 25 July, Cambridge University Botanic Gardens (Cambridge shire).
- A new winter record (21.2 °C), 26 February, Kew Gardens (London); the first time 20 °C has been reached in the UK in a winter month.
- A new December record (18.7 °C), 28 December, Achfary (Sutherland).
- A new February minimum record (13.9 °C), 23 February, Achnagart (Highland).

The building construction sector produces almost 30% of $CO_2$ emissions in the atmosphere and at least 60% of these are due to the use of the building during its lifetime, which shows the importance of the built environment in global warming and climate change [4]. The UK building sector accounts for approximately 3% of total electricity use and the UK supermarkets and similar organisations are responsible for 1% of the total UK GHG emissions [6]. Since climate change has a direct effect on the built environment, it is critical for the industry to quantify how the change in climate impacts the buildings. It is noted that it affects the functioning of a building by reducing winter heating demand and increasing summer cooling demand. This applies especially to supermarkets' operation as they are considered "high energy use intensity (EUI)" due to their increased refrigeration and lighting needs. For this reason, the scientific research community has been working on developing the science of building simulations to perform calculations based on future weather data based on atmospheric-ocean general circulation models (GCM) developed by Normal A. Philips to help predict climatic variations at a relatively high level of spatial resolution [7]. Reducing energy consumption and GHG emissions are one of the most important goals of European policies to achieve a sustainable and long-lasting future [8]. In 2018, as part of the "Clean Energy for all Europeans" package, a new target was set to cut energy consumption by at least 32.5% by 2030. Energy efficiency measures are increasingly recognized as a means not only to achieve a sustainable energy supply, cut greenhouse gas emissions, improve security of supply, and reduce import bills, but also to promote the EU's competitiveness. Energy efficiency is therefore a strategic priority for the Energy Union, and the EU promotes the principle of "energy efficiency first". The future policy framework for the post-2030 period is under discussion [9].

Figure 1 shows the global average surface temperature change from 2006 to 2100 under various representative concentration pathways (RCPs) including a stringent mitigation scenario (RCP2.6), two intermediate scenarios (RCP4.5 and RCP6.0), and one scenario with very high GHG emissions (RCP8.5) [10].

For the UK-based buildings, the most up-to-date and accurate climate projections are provided by UK Climate Projections (UKCP), which is a climate analysis tool and forms part of the Met Office Hadley Centre Climate Programme [11]. It assists to quantify the direct effect of climate change on the buildings by using future climatic projections.

These projections are available in three emission scenarios including high, medium, and low for both test reference years (TRY) and design summer years (DSY) [12].

TRY: A representative database of weather data for the 1 year duration is known as test reference year (TRY) or typical meteorological year (TMY). TMY is defined as a year that sums up all the climatic information characterizing a period of the mean life of the system [13]. The (TMY) data sets represent 1 year of hourly (8760) weather data values extracted from long-term (at a minimum, 10 years) data records. This data set is produced

from the US-based organization, The National Solar Radiation Database (NSRDB) and is an empirical method that involves selecting 12 months of data from the 30 year record available in the NSRDB based on five weather parameters: global horizontal irradiance (GHI), direct normal irradiation (DNI), dry bulb temperature, dew point temperature, and wind speed [14].

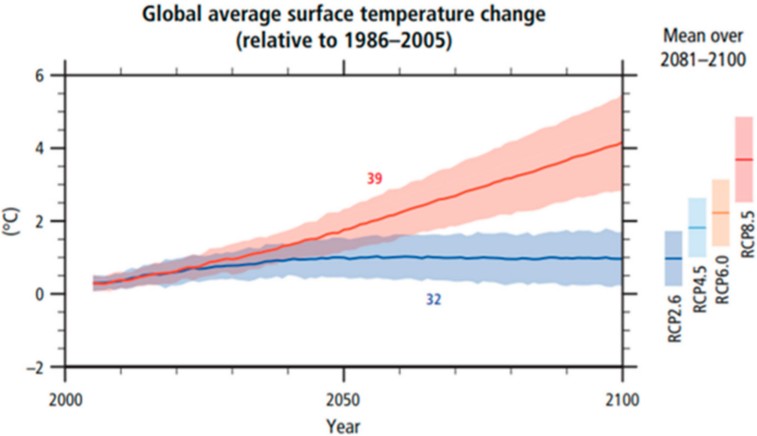

**Figure 1.** Global average surface temperature change from 2006 to 2100 [10].

DSY: The DSY is a single continuous year rather than a composite one made up from average months. The DSY is used for overheating analysis of the buildings.

Figure 2 shows annual $CO_2$ emissions for all the scenarios along with the various RCPs, which are the greenhouse gas concentration pathways.

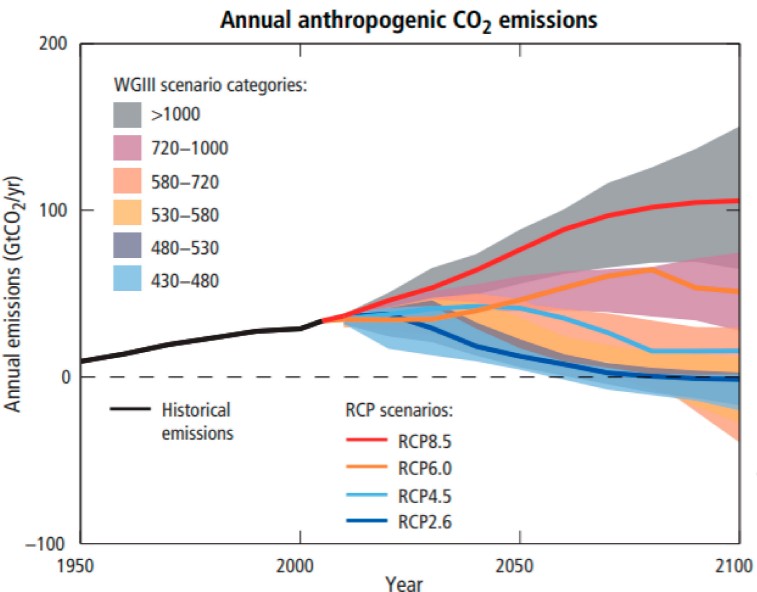

**Figure 2.** Annual carbon dioxide ($CO_2$) emissions in the representative concentration pathways (RCPs) and the associated scenario categories [10].

The future climate projections show that the mean surface temperature is projected to rise over the coming years under all the possible emission scenarios. The heatwaves will continue to happen more often, and last longer and extreme precipitation events will become more intense and frequent in many regions [10]. The increase in global mean surface temperature by the end of the 21st century (2081–2100) relative to 1986–2005 is likely to be 0.3 °C to 1.7 °C under RCP 2.6, 1.1 °C to 2.6 °C under RCP 4.5, 1.4 °C to 3.1 °C under RCP 6.0 and 2.6 °C to 4.8 °C under RCP 8.59 [15]. The Arctic region will

continue to warm more rapidly than the global mean (Table 1). These findings confirm the long-standing hypotheses that the warming of the climate system is unequivocal and will result in an increase in summer cooling demand and a reduction in winter heating demand. Thus, changes in weather conditions will impact building performance [16].

**Table 1.** Projected change in global mean surface temperature [10].

| | | 2046–2065 | | 2081–2100 | |
|---|---|---|---|---|---|
| | Scenario | Mean | Likely Range | Mean | Likely Range |
| Global mean surface temperature change (°C) | RCP 2.6 | 1.0 | 0.4 to 1.6 | 1.0 | 0.3 to 1.7 |
| | RCP 4.5 | 1.4 | 0.9 to 2.0 | 1.8 | 1.1 to 2.6 |
| | RCP 6.0 | 1.3 | 0.8 to 1.8 | 2.2 | 1.4 to 3.1 |
| | RCP 8.5 | 2.0 | 1.4 to2.6 | 3.7 | 2.6 to 4.8 |

These important findings prove that sensitivity to climate change is an important parameter in the functionality of buildings, especially in long-term predictions. Studies have suggested that for developed nations, such as "The Organisation for Economic Co-operation and Development" (OECD) countries, about 25–40% of anthropogenic greenhouse emissions will be related to buildings, and out of these, 40–95% will be caused by operational energy use with the rest being caused by construction and demolition of the building [17,18].

One of the earliest works to document the effect of climate change on building energy consumption was the report by the United States Congress by Loveland and Brown in 1989 presenting detailed research into five building types in six US cities, finding that the overall cooling demands would increase greatly irrespective of dominated load (internal or skin) [19]. In 2005, Gaterell and McEvoy produced a study to show the impact of climate change on detached dwellings' energy efficiency in the UK [20]. Another study was published in 2005 and 2008 to show the impact of climate change on the indoor environment, carbon dioxide emissions, and thermal mass, respectively [21,22]. In 2009, Lomas and Ji presented their study on natural ventilation in hospital wards using alternative weather projections [23]. Additionally, another study focusing on a specific building system of natural ventilation and focusing on wind prediction, using information from the UK Climate Projections 2009 (UKCP09), was presented in 2012 [24]. Another significant climate change study was published in 2012 that reported that the degree-day method and building simulation approach were the most popular study methods and that whether the reduction in heating demand would outweigh the increase in required cooling depended on the climate under consideration [25].

Despite all of these available materials and studies, there is still limited data regarding the impact of the future climatic conditions on the operational carbon emissions of the supermarket industry as much work has been done around other building types in the past and almost all of the aforementioned studies were published prior to the publication of UKCP 2009. UKCP09 builds on the success of its predecessors and uses state-of-the-art climatic science to provide a detailed future weather projection up to the year 2100 in the UK and globally. Based on these climatic projections, the resilience of buildings can be increased to future higher temperatures and the building's energy use can be assessed under future weather conditions. Furthermore, the future weather years were generated by climate scientists at Arup using a modified version of the "morphing" method developed in the research. The climate change projections used were the UK Government's UKCIP02 climate change scenarios [26]. The unchanged variables in the "morphing" technique are the present weather code (pwc) and wind direction; however, there are other factors such as atmospheric pressure, which undergoes a simple shift, the wind speed, specific humidity undergoes simple stretch and temperature undergoes shift and stretch whereas global solar irradiation undergoes weighted stretch. The derived variables include wet bulb temperature, cloud amount and diffused irradiation [27].

The Chartered Institution of Building Services Engineers (CIBSE) future weather files are available for building performance analysis for 14 UK locations including extra sites for London (Weather Centre (LWC); Gatwick (GTW)) for three time periods, the 2020s (2011–2040), the 2050s (2041–2070), and the 2080s (2071–2100). These weather files have been produced to assist academics and researchers in the use of weather and climate change information for building design and futureproofing of buildings. The data available are presented as TRY and DSY based on UKCP09 climate change scenarios and have the carbon emission scenarios of low, medium and high with varying levels of probabilities of 10th, 50th, and 90th percentiles [28]. The current TRY and DSY were morphed to incorporate the UKCP09 climate change scenarios of the time periods and the emission scenarios, helping to limit any uncertainties that could possibly affect the baseline weather data [29].

The building simulation and environmental performance software packages have been in use (and under constant development) for many decades and have the ability to evaluate a wide range of responses to the external stimuli [30]. The integrated modelling is defined as the best practice approach to building design as it allows the designers, architects, and engineers to link energy, the environment, and health by assessing the building's design, such as overheating analysis, assessment of internal conditions of the building (infiltration, ventilation, lightning gain, occupancy sensible and latent, equipment sensible and latent, and pollution generation), evaluation and enhancement of the building's thermal mass and evaluating alternate technologies (energy efficiency and renewable energy), and regulatory compliance and performance views [31,32].

This study makes use of a government-approved and validated thermal analysis building software package called thermal analysis simulation (TAS) by Environmental Design Solutions Limited (EDSL) to perform a series of simulations to quantify and predict the impact of changing future weather climatic conditions on a newly built LIDL baseline model in the UK. TAS EDSL is an elaborate software following the European technical standards as it follows all the technical memoranda of CIBSE and utilizes proven and empirical methods for estimating convective heat transfer from internal surfaces. TAS EDSL has several validations according to the European Standards (EN) such as ENISO13791: 2012/ EN ISO13792: 2012/ EN ISO15255: 2007/ EN ISO15265: 2007, International Commission on Illumination (CIE) 171:2006 (for daylighting calculations), as well as computational fluid dynamic (CFD) validation based on several critical factors [33]. This investigation will evaluate five key building performances: total annual energy consumption, annual building carbon emissions, annual electricity grid consumption, and cooling and heating demand based on the current and future CIBSE weather data set morphed from the UK Climate Projection 2009 weather information.

## 2. Methodology

### 2.1. Background

The objective of the study is to evaluate and predict the impact of future climatic conditions in a typical supermarket by considering a baseline LIDL supermarket store situated in UK. This is achieved through a series of simulations in a building services software package using the latest UK (CIBSE) current and UKCP09-based future TRY weather files.

TAS EDSL software version 9.5.0, which is a dynamic modelling package, assists in simulation of the thermal requirements for domestic and non-domestic dwellings. It offers a complete solution as a powerful modelling and simulation tool in the optimization of the building environment, energy performance, and occupant comfort [34]. TAS also provides the opportunity to combine the dynamic thermal simulation of the building with control functions over natural and mixed-mode ventilation [35]. A baseline model of a LIDL store is designed in the TAS EDSL software package and since the supermarket store is based in London (UK), the current and future CIBSE London TRY weather files are chosen for evaluation purposes.

## 2.2. Thermal Analysis Simulation (TAS EDSL) 3D Modelling

The TAS modelling contains AutoCAD architectural building drawings of the LIDL baseline supermarket store. The drawings consist of front, rear, and gable elevations. Along with it, it has the floor and roof plans to make it as accurate as possible. Figure 3a–d show the architectural drawings and their respective specification details.

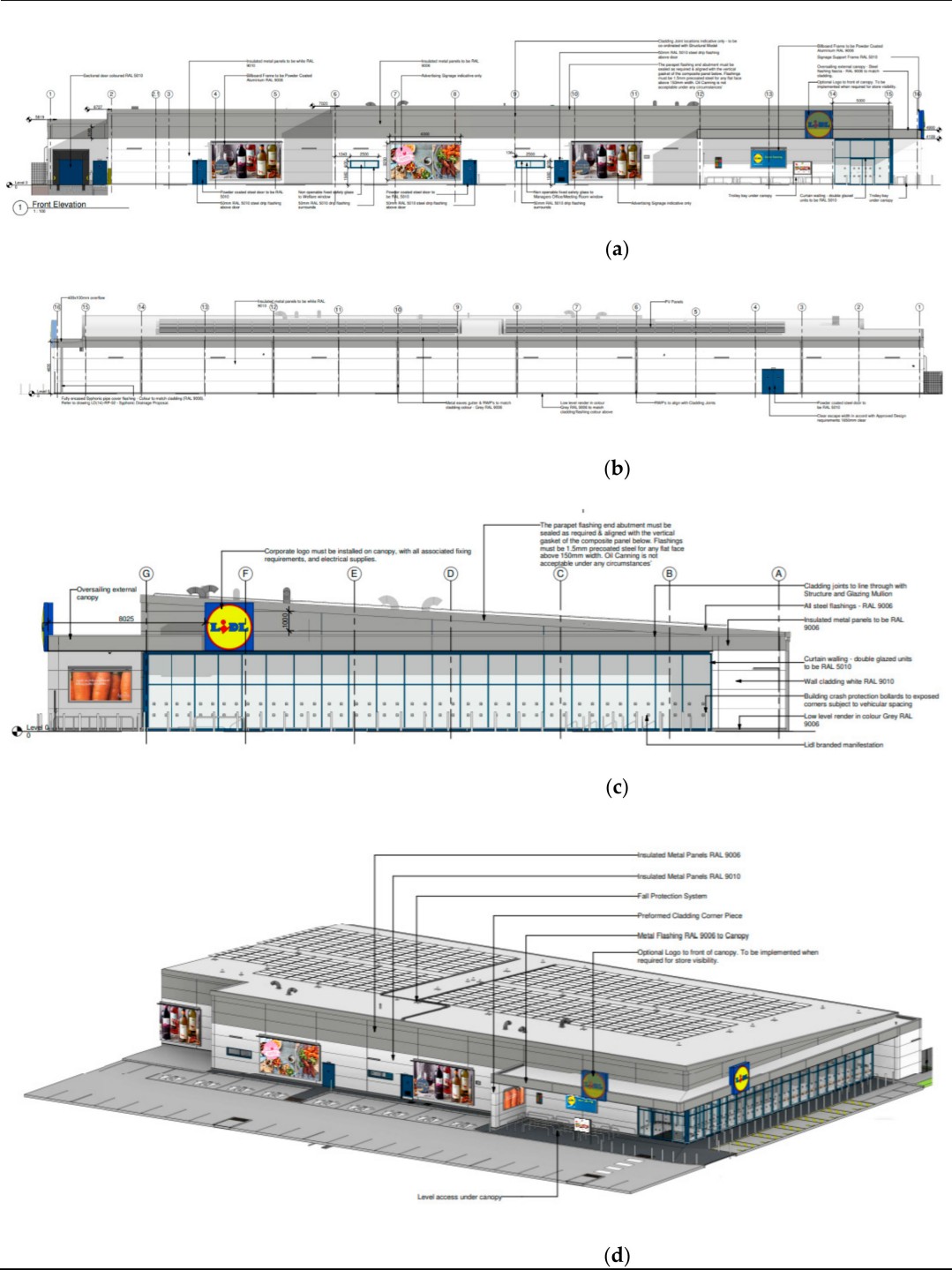

(a)

(b)

(c)

(d)

**Figure 3.** (**a**–**d**). LIDL Three Dimensional model building.

### 2.3. Modelling Process

The general arrangement architectural building drawings provide the exact measurements for the building height, sales area floor size, entrance lobby, bakery, warehouse, toilets, and other offices including Information Technology (IT) room, cash room, utility, meeting room, cloakroom, and welfare canteen. The total floor area of the building is around 2500 m². The National Calculation Methodology (NCM) standard calendar is used to reflect the operational hours of the supermarket. Construction materials are assigned individually to all the building elements of the store according to the LIDL specifications and internal conditions are applied to the individual zones. When designing the model building, certain precautions were taken to eliminate any miscalculations such that the floor level was measured from the ground floor at 0.0 metres and the heights of the wall were measured from the floor level to the directly below the finishing of the roof above. Furthermore, all the floor areas of the supermarket were divided into zones such as entrance lobby, store, sales area, warehouse, bakery, welfare canteen, cloakrooms, staff toilets (male and female), corridor, meeting room, utility, cash room, IT room, and customer Water Closet (WC) so that they can be assigned their respective internal conditions adhering to the national calculation method. As for the weathering profile, London TRY files will be used as the store is based in London making them the closest/most appropriate weather files [36]. These TRY files are used for predicting average energy consumption and compliance with the UK building regulations [37]. Moreover, the thermophysical characteristics of the building materials are summed up in a table, giving an insight into the construction modelling of a typical supermarket (Table 2).

**Table 2.** Construction details: specifications of thermophysical characteristics.

| Type | | Conductance (W/m2. $^0$C) | Solar Absorptance | | Emissivity | Time Constant | Construction Type |
|---|---|---|---|---|---|---|---|
| | | | External/Internal | | External/Internal | | |
| Wall | Cast Concrete wall | 0.974 | 0.700 | | 0.900 | 4.169 | Opaque |
| | Cavity wall | 0.25 | 0.700 | | 0.900 | 12.790 | Opaque |
| | Curtain Wall | 5.227 | 0.700 | | 0.900 | 0.0 | Opaque |
| | Metal Cladding Wall | 0.235 | 0.700 | | 0.900 | 0.0 | Opaque |
| | Steel Frame Wall | 0.379 | 0.700 | | 0.900 | 2.526 | Opaque |
| Frame | Uncoated glass, air-filled | 5.545 | 0.101 | 0.078 | 0.840 | 0.00 | Transparent |
| | Metal, thermal break & spacer | 59.116 | 0.00 | | 0.850 | 0.00 | Transparent |
| | Wood, thermal spacer | 7.89 | 0.00 | | 0.850 | 0.00 | Transparent |
| Floor | Ground Floor | 0.218 | 0.700 | | 0.900 | 156.820 | Opaque |
| Door | Insulated personal door | 0.94 | 0.700 | | 0.900 | 0.00 | Opaque |
| | Vehicle door | 2.0 | 0.700 | | 0.900 | 0.00 | Opaque |

### 2.4. Simulation Process

For the simulation process, the TAS modeller designs the thermal mass of a building and requires multiple performance parameters and assumptions to simulate the building without any errors and warnings. The various simulation parameters, including building summary, calendar, weather, building elements, zones, internal conditions, and schedule to simulate the building, are summarized in Tables 3 and 4.

**Table 3.** Simulation assumptions: building fabric specifications.

| Building Element | Calculated Area-Weighted Average U-values (W/m$^2$K) |
|---|---|
| Wall | 0.24 |
| Floor | 0.21 |
| Roof | 0.13 |
| Windows | 3.08 |
| Personnel doors | 1.32 |
| Vehicle access doors | 1.78 |
| High usage entrance doors | 3.34 |

**Table 4.** Simulation assumptions: building summary specifications.

| Calendar | NCM Standard |
|---|---|
| Air permeability | 4.0 m$^3$/h.m$^2$ @ 50Pa |
| Infiltration | 0.125 (ACH) |
| Fuel source | Grid supplied electricity |
| $CO_2$ factor | 0.519 kg/kWh |

*2.5. UK Building Regulation Studio 2013*

TAS EDSL v 9.5.0 comes fully equipped with a UK building regulation studio 2013. It helps in calculating Building Regulations United Kingdom part-L (BRUKL) and Energy Performance Certificate (EPC) documents in a clear and concise way by using the NCM for Energy Performance of Building Directive by Ministry of Housing, Communities and Local Government (DCLG) and generates compliance reports suggesting whether the building adheres to the part L2 building regulations. The dynamic modelling provides a detailed and comprehensive evaluation of the building with results that can be generated on an hourly basis and allows the comparison of information between the model building with a notional building to identify the potential compliance issues with the building design. Moreover, the studio generates valuable reports that include total annual energy consumption, annual electricity grid consumption, building emissions rate, and cooling/heating demand for this study [34].

In the baseline model, lighting control with specific auto presence detection, power efficacy and design room illuminance (lux) is applied to all the individual zones according to the LIDL specifications to reflect the actual store conditions. The model is also equipped with a number of air-sided configuration systems in place such as natural vent, sales area heating, ventilation, and air conditioning (HVAC), welfare mechanical ventilation with heat recovery (MVHR), welfare mechanical ventilation with heat recovery with air condition (MVHR with AC), air conditioning (AC) only, extract only, and storage MVHR with AC to supply the zones. Another part of the model is the design of heating and cooling configuration circuits with modifiable efficiency and fuel sources to serve all the required components. Lastly, the model has domestic hot water (DHW) circuit configuration to provide hot water to the required areas in the store such as the toilets and the welfare canteen.

*2.6. Future Weather Data Simulation Process*

The simulation covers the scenarios based on the current and future climate variables with different carbon emission scenarios (high, medium, and low) and for the time periods 2050s (2041–2070) and 2080s (2071–2100).

**3. Results and Discussion**

The analysis and model of baseline LIDL model supermarket based in London UK is presented in Figure 4a–c. These represent the results of the simulation modelling covering different sides of the building geometry.

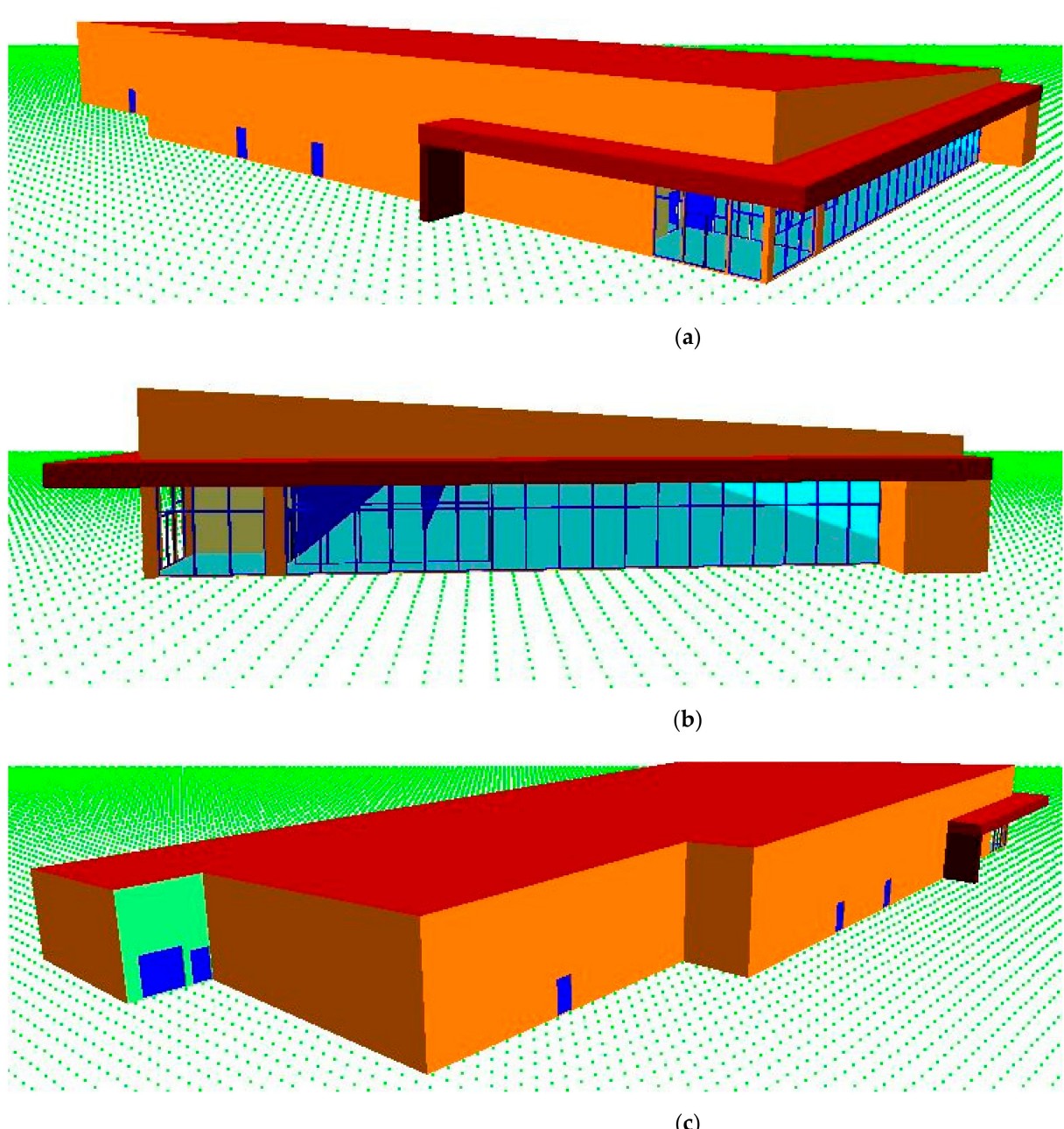

Figure 4. (**a**–**c**). Baseline model building geometry (thermal analysis simulation (TAS) software).

Table 5 shows the energy and $CO_2$ emissions summary from the Building Regulations United Kingdom part-L (BRUKL) output document. It compares the information of the actual building emissions to the notional building including heating and cooling demand, primary energy, and the total emissions.

**Table 5.** Energy and $CO_2$ emissions summary.

|  | Actual | Notional |
|---|---|---|
| Heating + cooling demand (MJ/m$^2$) | 594.54 | 599.9 |
| Primary energy (kWh/m$^2$) | 348.99 | 306.81 |
| Total emissions (kg/m$^2$) | 59 | 53.4 |

Another important parameter to compare is the external temperature of the building that will vary depending on the weather conditions. Figure 5 shows the minimum and maximum external temperatures as −3.2 °C and 30.7 °C, occurring on March 2 and July 14, respectively. All the information given in Figures 5 and 6 is used for further statistical analysis. Figure 6 shows a three dimensional visualization of the building's resultant temperature at peak external temperature (30.7 °C) on 14 July.

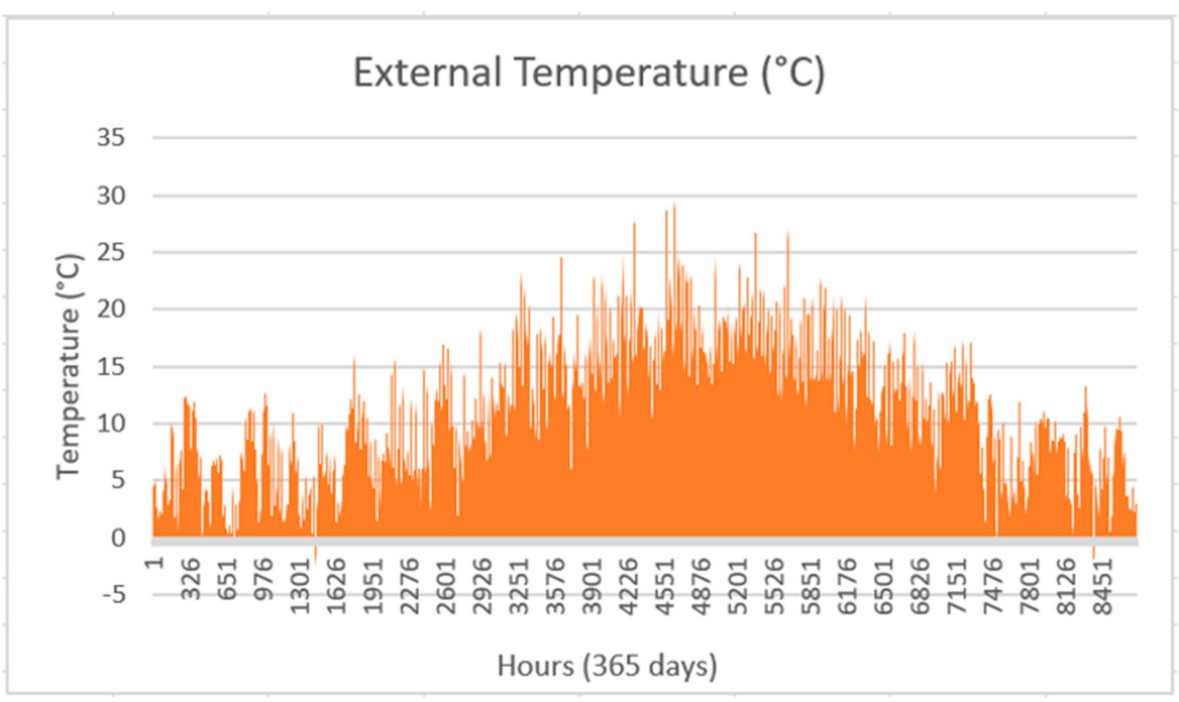

**Figure 5.** Annual hourly external temperature.

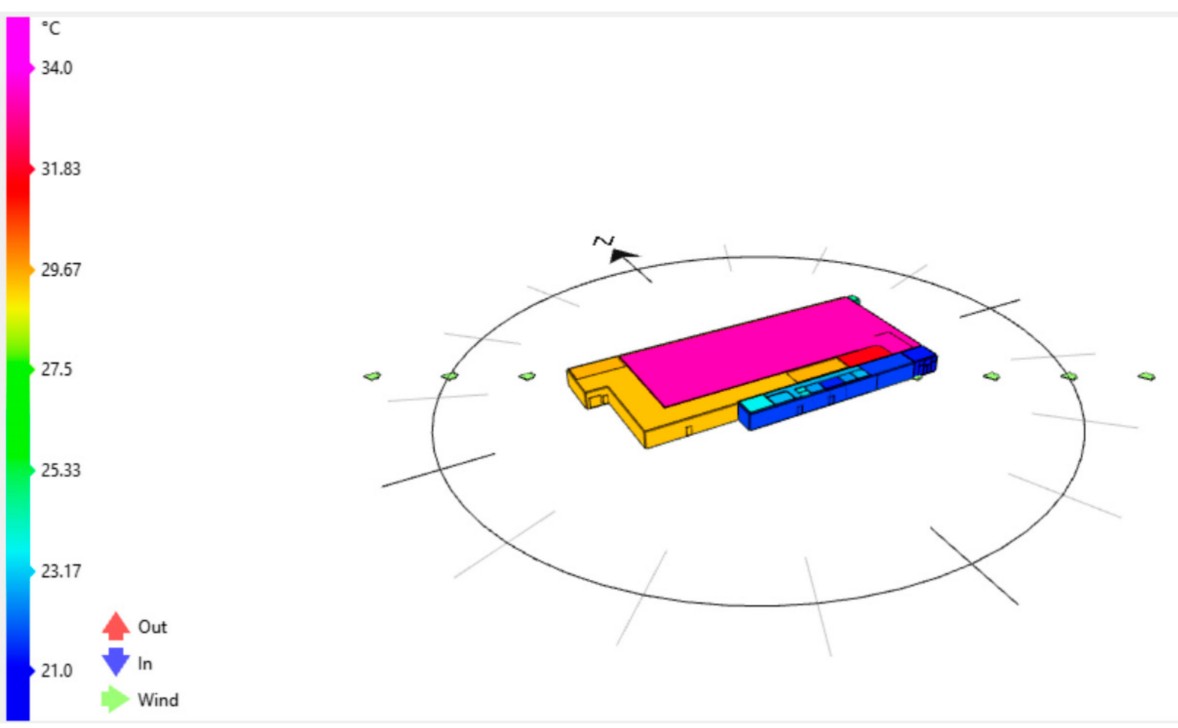

**Figure 6.** A 3D visualization of the building's resultant temperature at peak external temperature on 14 July.

### 3.1. Statistical Analysis of the Key Performance Indicators

The change in key performance indicators of the baseline LIDL supermarket model in current and future weather data is presented in Tables 6–15 and Figures 7–11. It shows the percentage variations of the building performance indicators using the future weather data timeline scenarios when compared to the current weather data.

**Table 6.** Annual energy consumption variation comparison under the 2050s percentile.

| | Total Annual Energy Consumption (kWh/m$^2$) | | | | | | |
|---|---|---|---|---|---|---|---|
| | | The 2050s | | | | | |
| Baseline LIDL model | Current (kWh/m$^2$) | Med (10th) %Inc | Med (50th) %Inc | Med (90th) %Inc | High (10th) %Inc | High (50th) %Inc | High (90th) %Inc |
| | 98.63 | 1.80 | 4.12 | 7.01 | 1.46 | 3.80 | 6.45 |

**Table 7.** Annual energy consumption variation comparison under the 2080s percentile.

| | Total Annual Energy Consumption (kWh/m$^2$) | | | | | | | | | |
|---|---|---|---|---|---|---|---|---|---|---|
| | | The 2080s | | | | | | | | |
| Baseline LIDL model | Current (kWh/m$^2$) | Low (10th) %Inc | Low (50th) %Inc | Low (90th) %Inc | Med (10th) %Inc | Med (50th) %Inc | Med (90th) %Inc | High (10th) %Inc | High (50th) %Inc | High (90th) %Inc |
| | 98.63 | 2.92 | 6.48 | 11.05 | 3.96 | 8.22 | 14.07 | 5.14 | 10.36 | 17.68 |

**Table 8.** Annual $CO_2$ emissions variation comparison under the 2050s percentile emissions.

| | Annual $CO_2$ Emissions Comparison (kgCO$_2$/m$^2$) | | | | | | |
|---|---|---|---|---|---|---|---|
| | | The 2050s | | | | | |
| Baseline LIDL model | Current (kgCO$_2$/m$^2$) | Med (10th) % Inc | Med (50th) % Inc | Med (90th) % Inc | High (10th) % Inc | High (50th) % Inc | High (90th) % Inc |
| | 51.29 | 1.60 | 3.90 | 6.80 | 1.25 | 3.61 | 6.24 |

**Table 9.** Annual $CO_2$ emissions variation comparison under the 2080s percentile emissions.

| | Annual $CO_2$ Emissions Comparison (kgCO$_2$/m$^2$) | | | | | | | | | |
|---|---|---|---|---|---|---|---|---|---|---|
| | | The 2080s | | | | | | | | |
| Baseline LIDL model | Current (kgCO$_2$/m$^2$) | Low (10th) %Inc | Low (50th) %Inc | Low (90th) %Inc | Med (10th) %Inc | Med (50th) %Inc | Med (90th) %Inc | High (10th) %Inc | High (50th) %Inc | High (90th) %Inc |
| | 51.29 | 2.71 | 6.26 | 10.84 | 3.76 | 8.01 | 13.84 | 4.93 | 10.14 | 17.45 |

**Table 10.** Annual electricity energy variation comparison for the 2050s percentile.

| | Annual Electricity Energy Comparison (kWh/m$^2$) | | | | | | |
|---|---|---|---|---|---|---|---|
| | | The 2050s | | | | | |
| Baseline LIDL model | Current (kWh/m$^2$) | Med (10th) %Inc | Med (50th) %Inc | Med (90th) %Inc | High (10th) %Inc | High (50th) %Inc | High (90th) %Inc |
| | 303.39 | 1.61 | 3.91 | 6.80 | 1.26 | 3.60 | 6.24 |

**Table 11.** Annual electricity energy variation comparison for the 2080s percentile.

| | Annual Electricity Energy Comparison (kWh/m$^2$) | | | | | | | | |
| --- | --- | --- | --- | --- | --- | --- | --- | --- | --- |
| | | | | | The 2080s | | | | |
| Baseline LIDL model | Current (kWh/m$^2$) | Low (10th) %Inc | Low (50th) %Inc | Low (90th) %Inc | Med (10th) %Inc | Med (50th) %Inc | Med (90th) %Inc | High (10th) %Inc | High (50th) %Inc | High (90th) %Inc |
| | 303.39 | 2.72 | 6.27 | 10.83 | 3.76 | 8.01 | 13.85 | 4.93 | 10.14 | 17.45 |

**Table 12.** Annual cooling energy consumption variation comparison for the 2050s percentile.

| | Annual Cooling Energy Consumption comparison (kWh/m2) | | | | | | |
| --- | --- | --- | --- | --- | --- | --- | --- |
| | | | | The 2050s | | | |
| Baseline LIDL model | Current (kWh/m$^2$) | Med (10th) %Inc | Med (50th) %Inc | Med (90th) %Inc | High (10th) %Inc | High (50th) %Inc | High (90th) %Inc |
| | 53.74 | 3.00 | 7.29 | 12.67 | 2.36 | 6.70 | 11.63 |

**Table 13.** Annual cooling energy variation comparison for the 2080s percentile.

| | Annual Cooling Energy Consumption Comparison (kWh/m$^2$) | | | | | | | | | |
| --- | --- | --- | --- | --- | --- | --- | --- | --- | --- | --- |
| | | | | | | The 2080s | | | | |
| Baseline LIDL model | Current (kWh/m$^2$) | Low (10th) %Inc | Low (50th) %Inc | Low (90th) %Inc | Med (10th) %Inc | Med (50th) %Inc | Med (90th) %Inc | High (10th) %Inc | High (50th) %Inc | High (90th) %Inc |
| | 53.74 | 5.58 | 11.69 | 20.15 | 7.02 | 14.91 | 25.72 | 9.17 | 18.85 | 32.38 |

**Table 14.** Annual heating demand variation comparison for the 2050s percentile.

| | Annual Heating Energy Consumption Comparison (kWh/m$^2$) | | | | | | |
| --- | --- | --- | --- | --- | --- | --- | --- |
| | | | | The 2050s | | | |
| Baseline LIDL model | Current (kWh/m$^2$) | Med (10th) %Dec | Med (50th) %Dec | Med (90th) %Dec | High (10th) %Dec | High (50th) %Dec | High (90th) %Dec |
| | 0.19 | 15.79 | 31.58 | 47.37 | 15.79 | 31.58 | 47.37 |

**Table 15.** Annual heating demand variation comparison for the 2080s percentile.

| | Annual Heating Energy Consumption Comparison (kWh/m$^2$) | | | | | | | | | |
| --- | --- | --- | --- | --- | --- | --- | --- | --- | --- | --- |
| | | | | | | The 2080s | | | | |
| Baseline LIDL model | Current (kWh/m$^2$) | Low (10th) %Dec | Low (50th) %Dec | Low (90th) %Dec | Med (10th) %Dec | Med (50th) %Dec | Med (90th) %Dec | High (10th) %Dec | High (50th) %Dec | High (90th) %Dec |
| | 0.19 | 26.32 | 47.37 | 68.42 | 26.32 | 52.63 | 73.68 | 31.58 | 63.16 | 84.21 |

3.1.1. Total Annual Energy Consumption Variation

Tables 6 and 7 show the annual energy consumption for current and future climatic projections for the 2050s period, for medium and high emission scenarios, and for 10th, 50th, and 90th percentiles, and similarly for the 2080s period, they provide the energy consumption for the three emission scenarios of low, medium and high, for the 10th, 50th, and 90th percentile. All the predicted scenarios show that there is a constant gradual increase in energy consumption over the years, irrespective of any scenario or percentile chosen.

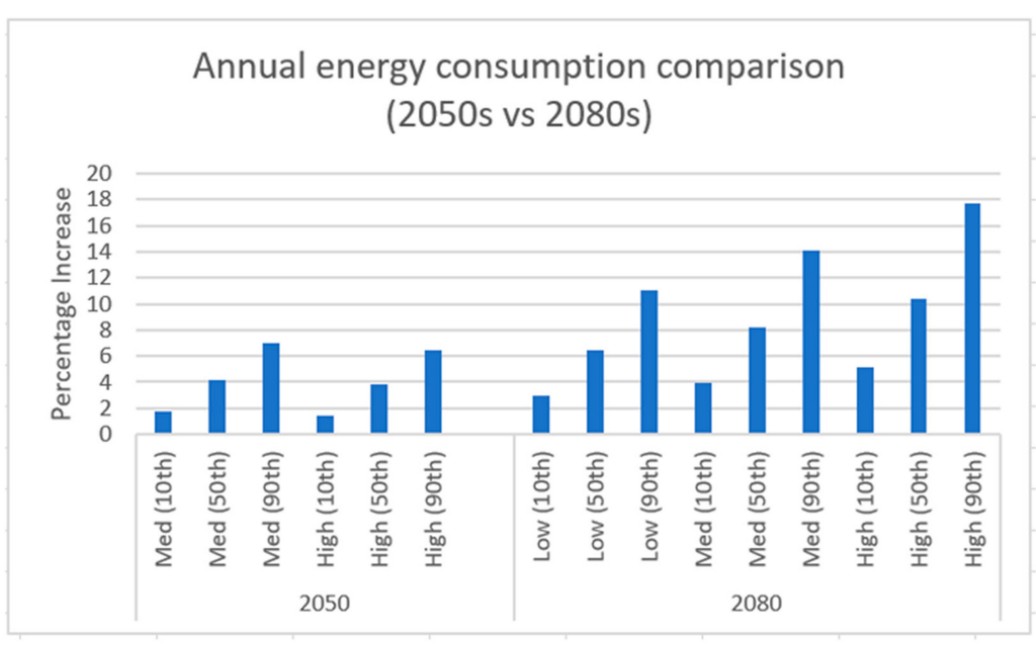

**Figure 7.** Annual energy consumption comparison (2050s vs. 2080s).

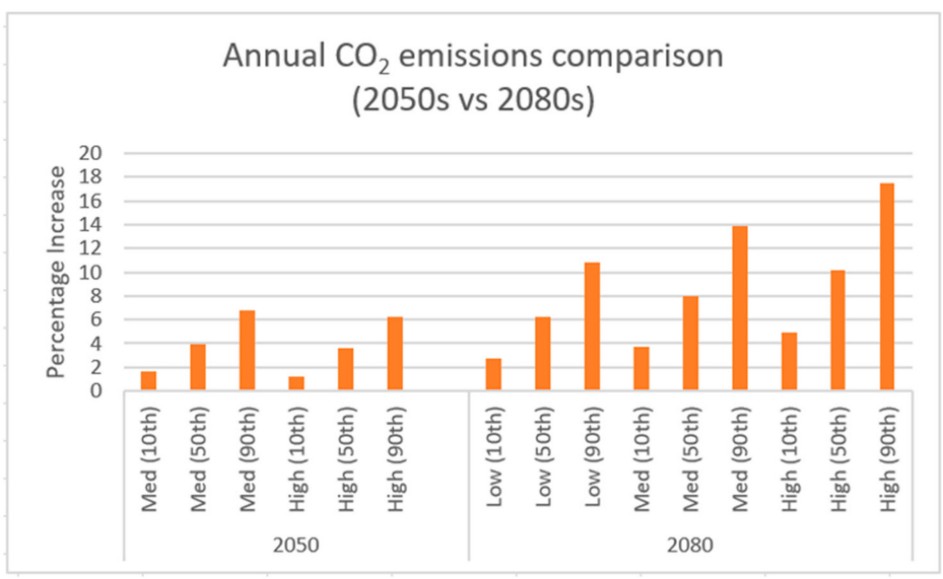

**Figure 8.** Annual $CO_2$ emissions comparison (2050s vs 2080s).

This trend is observed in all the emission scenarios with a peak increase of 7.01% in the 2050s medium (90th) percentile scenario and a 17.68% increase in the 2080s high (90th) percentile scenario, respectively. This rise in energy consumption is in accordance with the range of annual average temperature variation predicted by the Intergovernmental Panel on Climate Change (IPCC) scenarios, showing a gradual increase in the temperature over time. The increased energy consumption over the years is attributed to the increased cooling demand in the face of increasing climatic temperature.

3.1.2. Total Carbon Dioxide ($CO_2$) Emissions Variation

Tables 8 and 9 show the annual building $CO_2$ emissions for current and future climatic projections for the 2050s period, for medium and high emission scenarios and 10th, 50th and 90th percentiles, and similarly for the 2080s period, they provide the annual building $CO_2$ emissions for the low, medium, and high emission scenarios, and 10th, 50th and 90th

percentiles. All the predicted scenarios show that there is a constant gradual increase in carbon dioxide emissions over the years, irrespective of any scenario or percentile chosen.

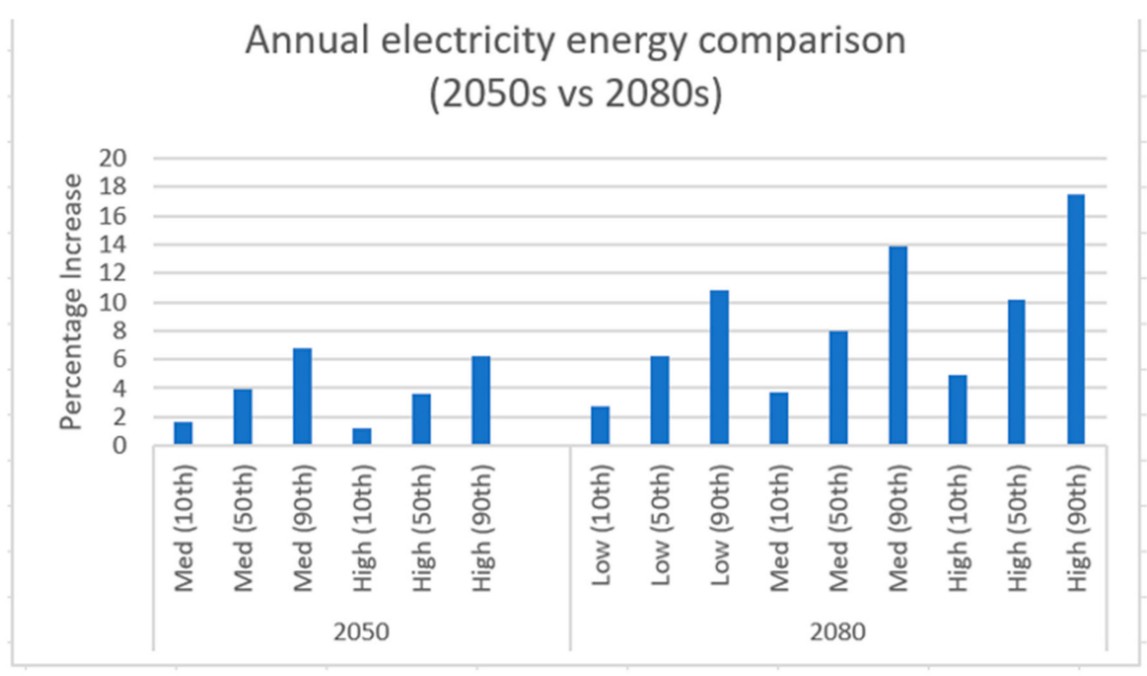

**Figure 9.** Annual electricity energy comparison (2050s vs. 2080s).

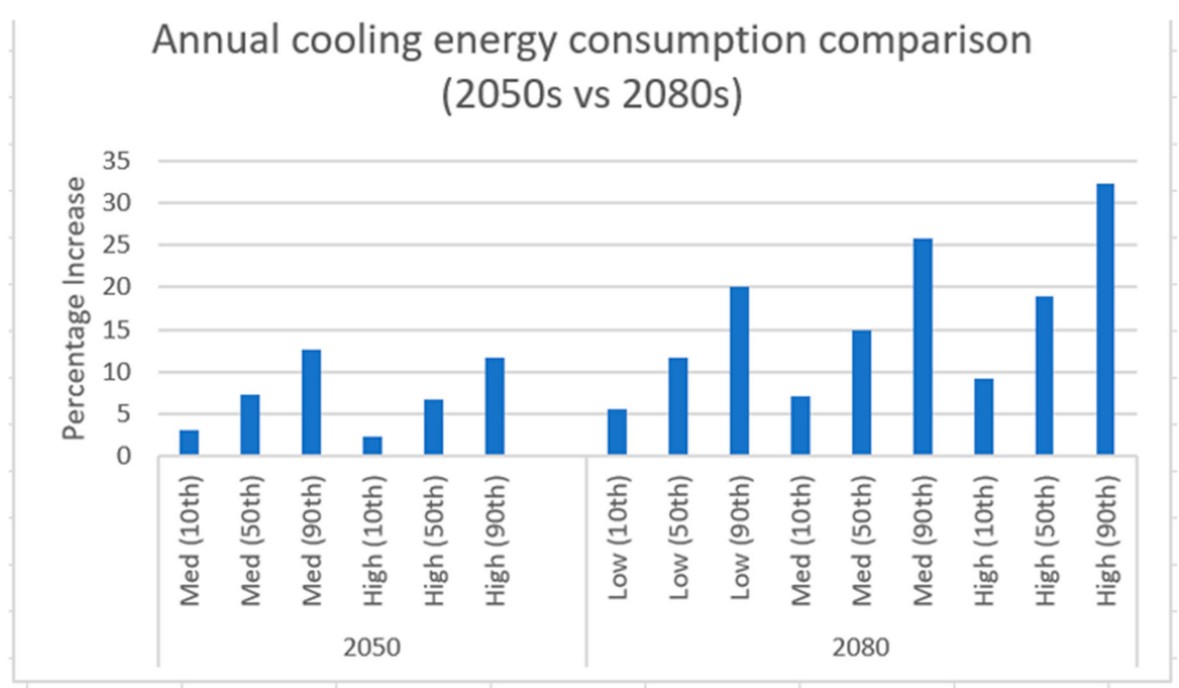

**Figure 10.** Annual cooling energy consumption comparison (2050s vs. 2080s).

This trend is observed in all the emission scenarios with a peak increase of 6.80% in the 2050s medium (90th) percentile scenario and a 17.45% increase in the 2080s high (90th) percentile scenario, respectively. This rise in carbon dioxide emissions is in accordance with the range of annual average temperature variation predicted by the IPCC scenarios, showing a gradual increase in the temperature over time. The increased emissions over the years are attributed to the increased cooling demand, making use of more electricity to match the increased energy demand in the face of increasing climatic temperature.

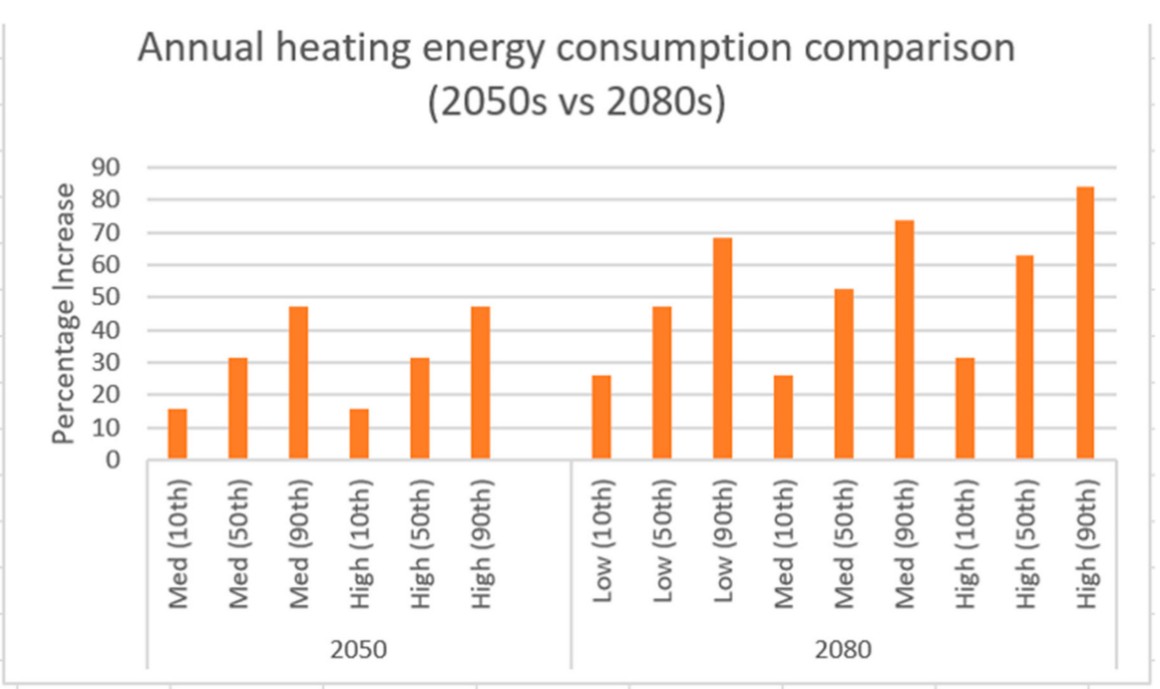

**Figure 11.** Annual heating energy consumption comparison (2050s vs. 2080s).

### 3.1.3. Annual Electricity Grid Comparison Analysis

Tables 10 and 11 show the annual electricity consumption for current and future climatic projections for the 2050s period, for medium and high emission scenarios, and for 10th, 50th and 90th percentiles, and similarly for the 2080s period, they provide the annual building electricity consumption for the low, medium, and high emission scenarios, and 10th, 50th and 90th percentiles. All the predicted scenarios show that there is a constant gradual increase in energy consumption over the years, irrespective of any scenario or percentile chosen.

This trend is observed in all the emission scenarios with a peak increase of 6.80% in the 2050s medium (90th) percentile scenario and a 17.45% increase in the 2080s high (90th) percentile scenario, respectively. This rise in the annual electricity consumption is in accordance with the range of annual average temperature variation predicted by the IPCC scenarios, showing a gradual increase in the temperature over time. The increased emissions over the years are attributed to the increased cooling energy demand as more electricity is used for matching the increased cooling demand in the supermarket.

### 3.1.4. Percentage of Cooling Demand Variation

Tables 12 and 13 show the annual cooling energy consumption for current and future climatic projections for the 2050s period, for medium and high emission scenarios and 10th, 50th and 90th percentiles, and similarly for the 2080s period, they provide the annual building cooling energy consumption for the low, medium and high emission scenarios, and 10th, 50th and 90th percentiles. All the predicted scenarios show that there is a constant gradual increase in the cooling energy consumption over the years, irrespective of any scenario or percentile chosen.

This trend is observed in all the emission scenarios with a peak increase of 12.67% in the 2050s medium (90th) percentile scenario and a 32.38% increase in the 2080s high (90th) percentile scenario, respectively. This rise in the annual electricity consumption is in accordance with the range of annual average temperature variation predicted by the IPCC scenarios, showing a gradual increase in the temperature over time. The increased cooling consumption over the years is attributed to the increasing external temperature, and thus increasing the need for cooling in the supermarket.

### 3.1.5. Percentage of Heating Demand Reduction

Tables 14 and 15 show the annual heating consumption for current and future climatic projections for the 2050s period, for medium and high emission scenarios and 10th, 50th and 90th percentiles, and similarly for the 2080s period, they provide the annual building heating demand for the low, medium and high emission scenarios, and 10th, 50th and 90th percentiles. All the predicted scenarios show that there is a constant gradual decrease in energy consumption over the years, irrespective of any scenario or percentile chosen.

This trend is observed in all the emission scenarios with a peak reduction of 47.37% in the 2050s medium (90th) percentile/high (90th) percentile scenario and an 84.21% reduction in the 2080s high (90th) percentile scenario, respectively. This fall in the annual heating consumption is in accordance with the range of annual average temperature variation predicted by the IPCC scenarios, showing a gradual increase in the temperature over time. The reduced heating consumption over the years is attributed to the increase in climatic temperature influencing the heating demand to be minimized.

The study, therefore, points to the fact that an increase in future temperature due to climatic variation would obviously have a significant declining impact on heating demand and conversely an increasing effect on the cooling demand in the supermarket industry.

### 3.2. *Analysis and Comparison of Significant Parameters under the Worst-Case Scenario*

To understand the far-reaching effects of the climatic variation on the heating, cooling and other significant parameters of the supermarket industry, simulations were run for the current weather data scenario and the worst-case scenario of the 2080s high (90th) percentile. The results are presented in Figures 12–23, showing the variations as graphs of temperature and loads and total load profile for the building between the three warmest months of the year (June 1 to August 31) for the current weather and for the 2080s high (90th) percentile whereas for the heating profile, the three coldest months have been used (January 01 to April 04), all of these simulations were run using the TRY weather files.

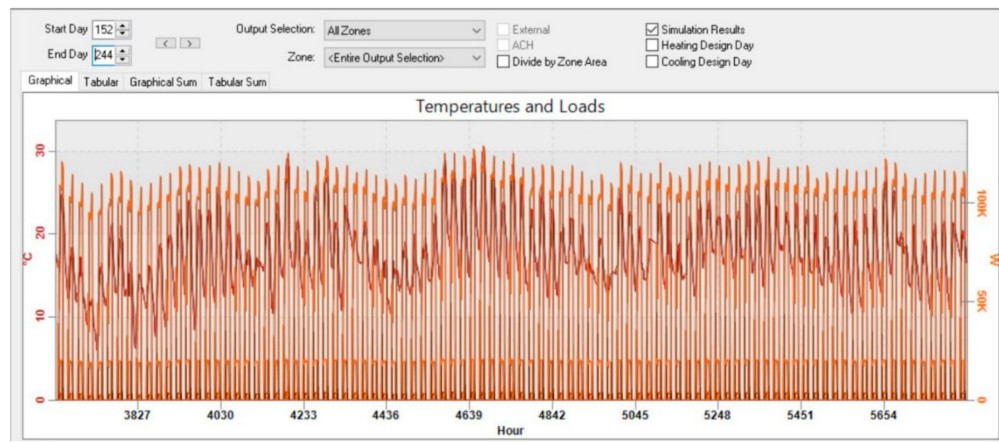

**Figure 12.** External temperature—current weather data.

### 3.2.1. Percentage of Heating Demand Reduction

An evaluation of the variation of temperature and load analysis is presented in Figures 12 and 13. The future predicted climate variation of the 2080s clearly shows a high upsurge in the temperature as compared to the current weather data. The comparison of the external temperature for the current weather data and the worst-case scenario of the 2080s high (90th) percentile shows that the external temperature ranges from 6 °C to 30.7 °C with relatively few periods going above the 30 °C mark for the current weather data; however, for the worst-case scenario, the external temperature ranges from 9.4 °C to 36.6 °C with 55 occurrences of above the 30 °C mark for the specified period of analysis, respectively.

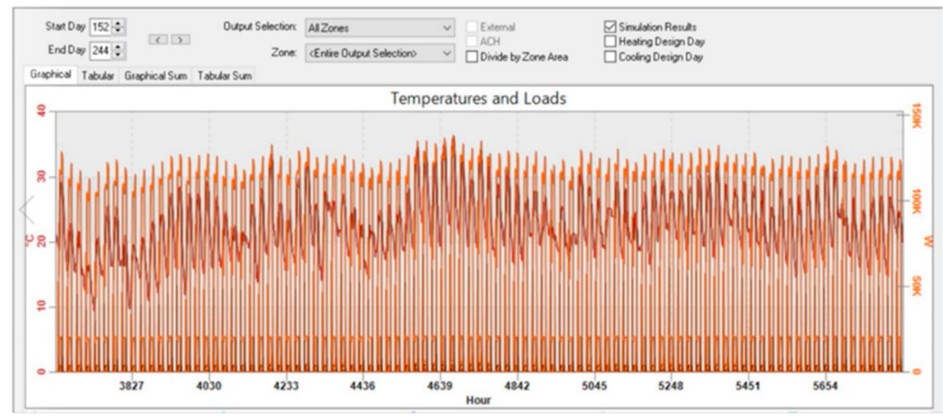

**Figure 13.** External temperature—the 2080s high (90th) percentile.

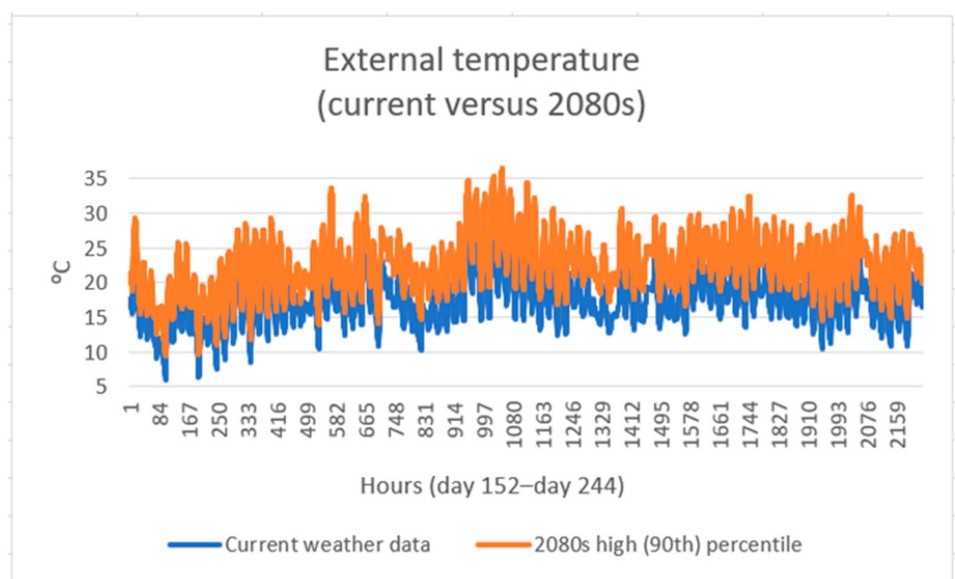

**Figure 14.** External temperature comparison: the current weather data versus the 2080s scenario (1 June–31 August).

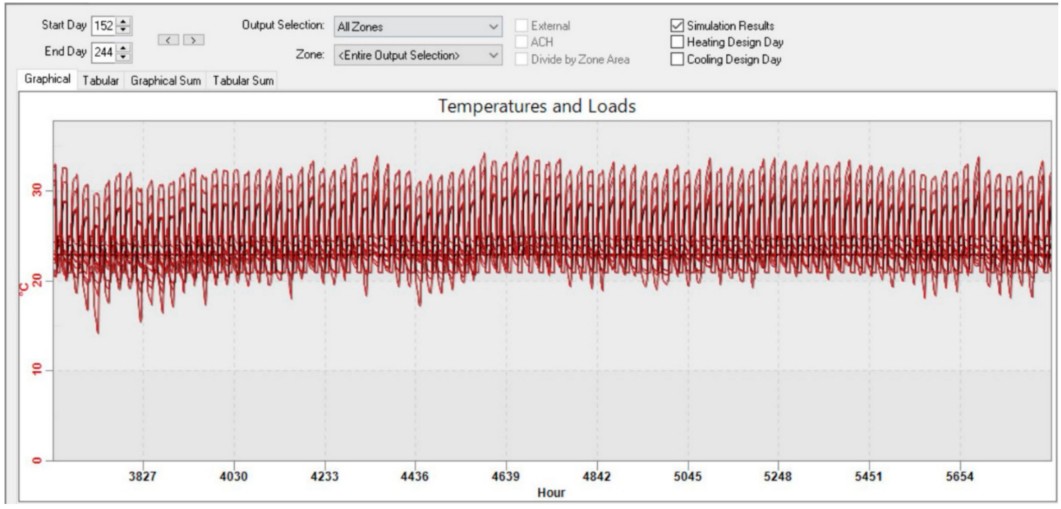

**Figure 15.** Dry bulb temperature—current weather data.

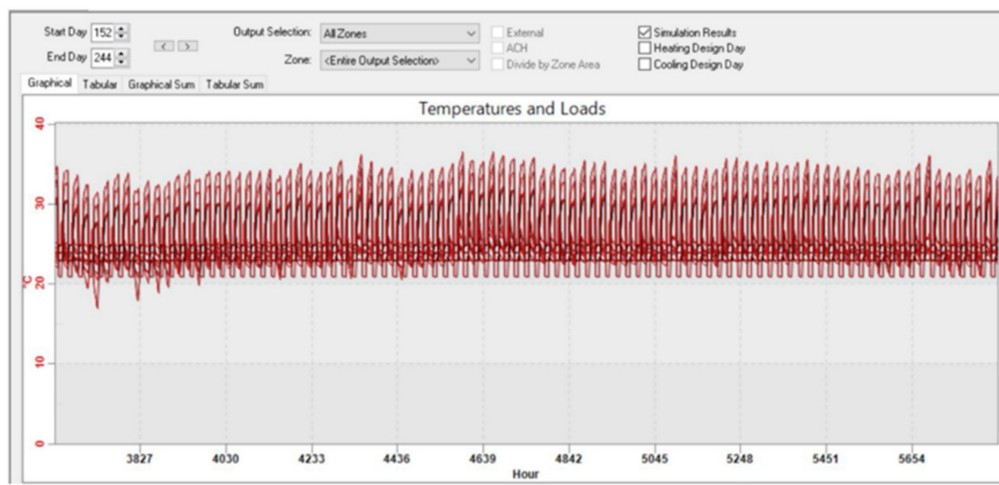

**Figure 16.** Dry bulb temperature—the 2080s high (90th) percentile data.

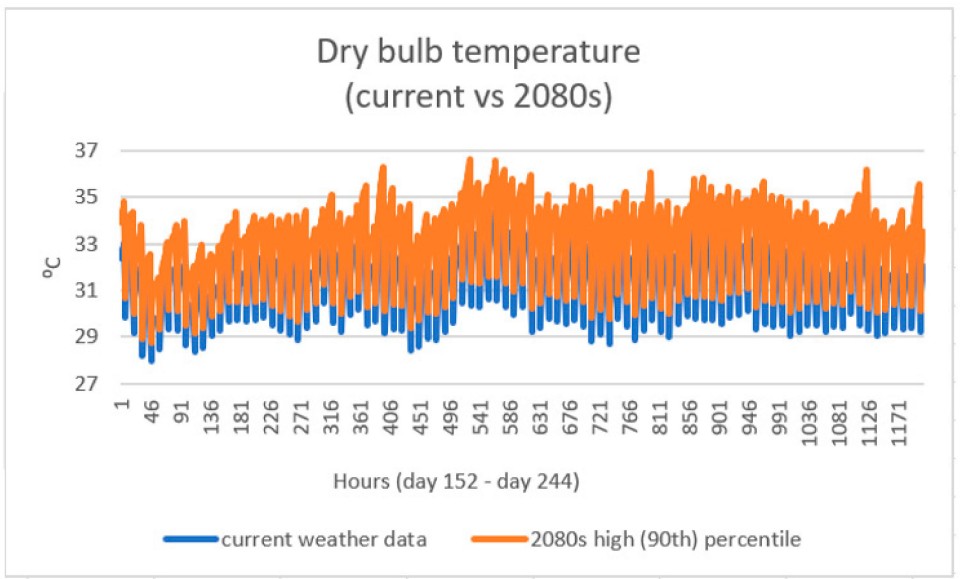

**Figure 17.** Dry bulb temperature comparison: the current weather data versus the 2080s scenario (1 June–31 August).

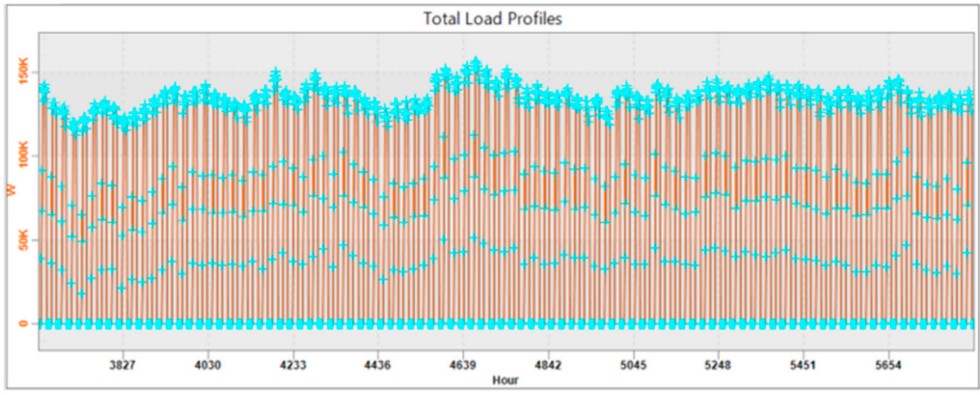

**Figure 18.** Cooling load profile—current weather data.

A comparison of the two external temperatures between the current weather data set and the 2080s high (90th) percentile data over the time period of June 1 and August 31 is

shown in Figure 14. It clearly indicates that the temperatures in the latter weather data set rise quickly with a peak increase of 19.22% between the two climatic variations.

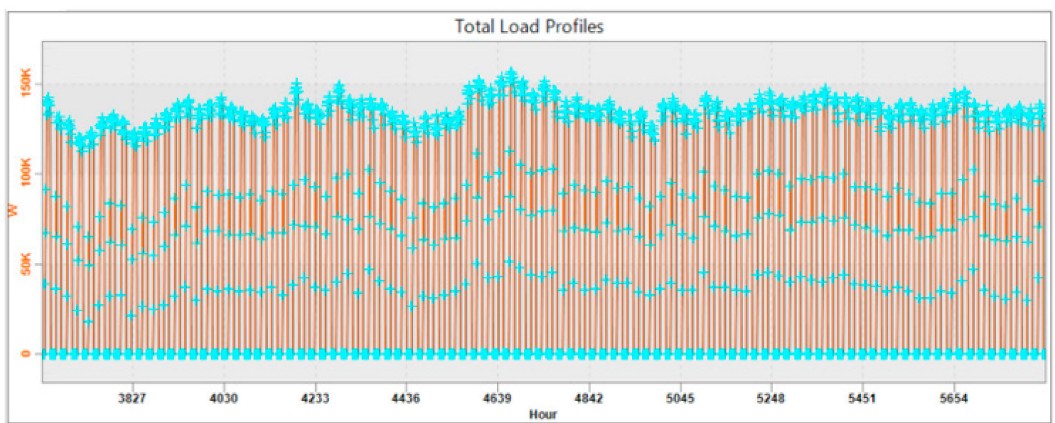

**Figure 19.** Cooling load profile—the 2080s high (90th) percentile.

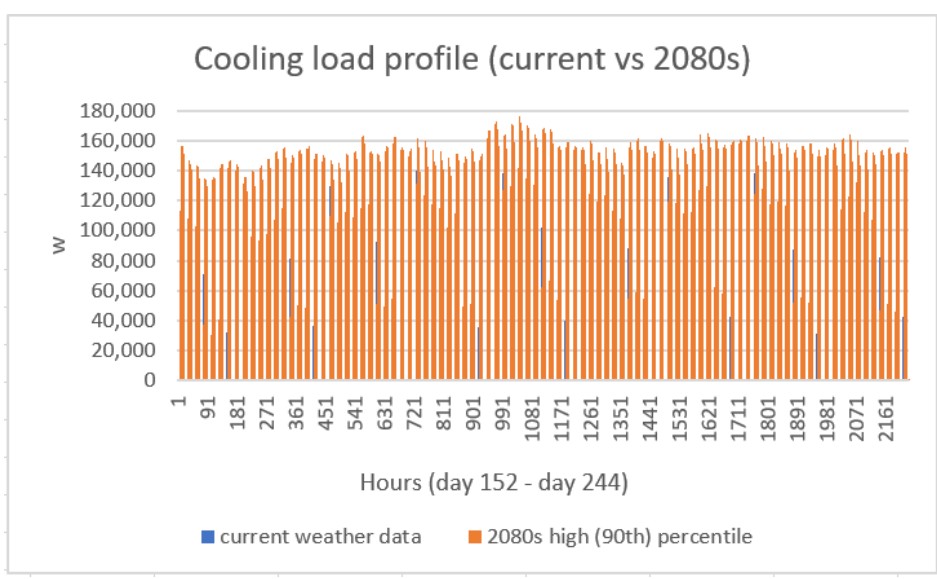

**Figure 20.** Cooling load profile comparison: the current weather data versus the 2080s scenario (1 June–31 August).

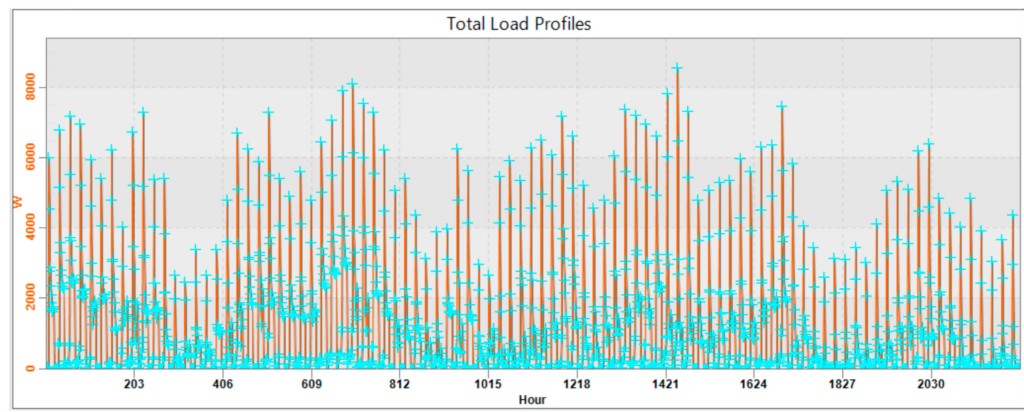

**Figure 21.** Heating load profile—current weather data.

### 3.2.2. Analysis of Dry Bulb Temperature

An evaluation of the variation of temperature and load analysis is presented in Figures 15 and 16. The future predicted climate variation of the 2080s clearly shows a high upsurge in the temperature as compared to the current weather data. The comparison of the temperature load analysis for the current weather data and the worst-case scenario of the 2080s high (90th) percentile shows that dry bulb temperature ranges from 12.0 °C to 34.44 °C with relatively few periods going above the 30 °C mark for the current weather data; however, for the worst-case scenario, the external temperature ranges from 12.0 °C to 36.62 °C with the time periods going above the 30 °C mark over 1.5 times more for the specified period of analysis, respectively.

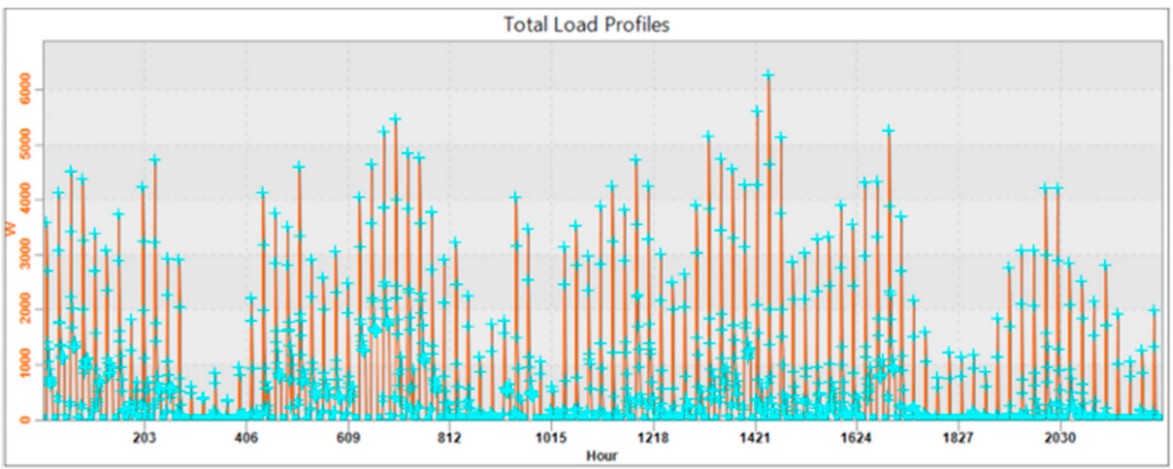

**Figure 22.** Heating load profile—the 2080s high (90th) percentile.

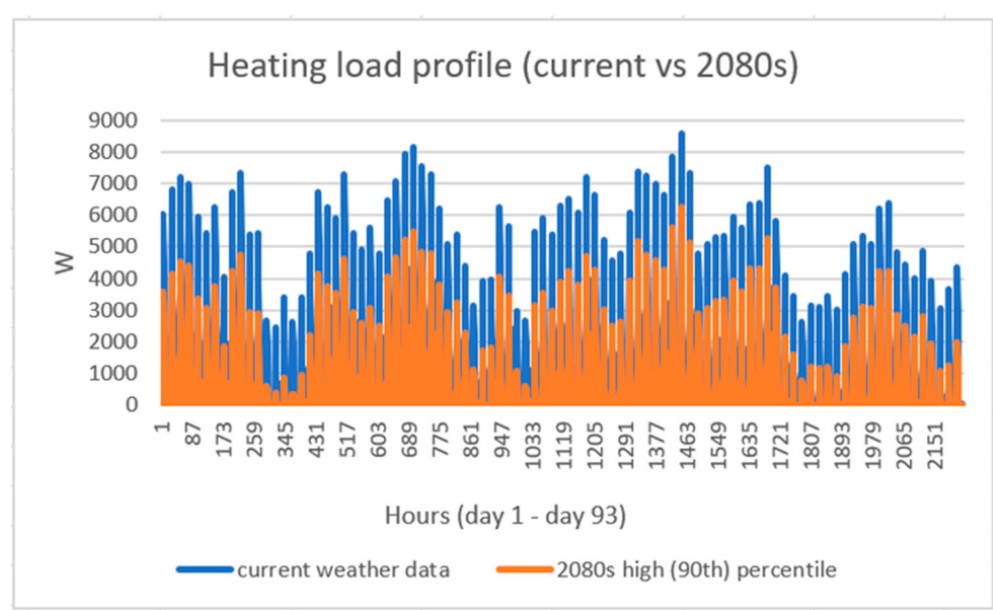

**Figure 23.** Heating load profile comparison: the current weather data versus the 2080s scenario (1 January–4 April).

A comparison of the two dry bulb temperatures between the current weather data set and the 2080s high (90th) percentile data over the time period of June 1 and August 31 is shown in Figure 17, highlighting the differences between the two weather data sets. It indicates that the dry bulb temperatures in the latter weather data set rise quickly with a peak increase of 6.33% between the two climatic variations.

### 3.2.3. Analysis and Comparison of Cooling Load Profile

An evaluation of the variation of the total load profile is presented in Figures 18 and 19. The future predicted climate variation of the 2080s clearly shows a high upsurge in the cooling load as compared to the current weather data.

The comparison of the total load profile analysis for the current weather data and the worst-case scenario of the 2080s high (90th) percentile shows that the cooling load profile ranges from 369.90 W to 156,787 W with relatively few periods going above the 150,000 W mark for the current weather data; however, for the worst-case scenario, cooling load profile ranges from 405.59 W to 176,573 W with the periods going above the 150,000 W mark over 25 times more for the specified period of analysis, respectively.

A comparison of the two cooling load profiles between the current weather data set and the 2080s high (90th) percentile data over the time period of June 1 and August 31 is shown in Figure 20. It indicates that the cooling load in the latter weather data set rises quickly with a peak increase of 12.62% between the two climatic variations.

### 3.3. Analysis and Comparison of Significant Parameters under the Worst-Case Scenario

An evaluation of the variation of the total load profile is presented in Figures 21 and 22. The future predicted climate variation of the 2080s clearly shows a significant reduction in the heating load as compared to the current weather data.

The comparison of the total load profile analysis for the current weather data and the worst-case scenario of the 2080s high (90th) percentile shows that the heating load profile ranges from 0 W to 6268.19 W with relatively few periods going above the 5000 W mark for the 2080s high (90th) percentile weather data; however, for the current weather data, the heating load profile ranges from 0 W to 8569.84 W with the time periods going above the 5000 W mark over 11 times more for the specified period of analysis, respectively.

A comparison of the two heating load profiles between the current weather data set and the 2080s high (90th) percentile data over the time period of 1 January and 4 April is shown in Figure 23. It indicates that the heating load in the latter weather data set declines steeply with a reduction of 36.72% between the two climatic variations.

It reinforces the idea that, due to the varying nature of the future weather and an increase in the overall temperature, the heating required in the year 2080 is significantly less as compared to the current weather data, which goes on to show that climate change is on the rise and the supermarket building's heating needs will change accordingly.

### 4. Conclusions

The study investigated the variability of future climatic conditions on a typical UK supermarket. The analysis of simulation results leads to the prediction of consistent inclination of annual building energy consumption, building emission rate, annual building electricity consumption, cooling demand, and a declining trend in heating demand over the different timelines of the 2050s and the 2080s used in the simulation.

The peak percentage increase for the annual energy consumption for current and future weather data set observed was 7.01 and 6.45 for the 2050s medium (90th) percentile and high (90th) percentile, respectively, and 11.05, 14.07, and 17.68 for the 2080s low (90th) percentile, medium (90th) percentile and high (90th) percentile, respectively. A similar rising trend in the case of annual $CO_2$ emissions was observed where the peak increase percentage was 6.80 and 6.24 for the 2050s medium (90th) percentile and high (90th) percentile, respectively, and 10.84, 13.84, and 17.45 for the 2080s low (90th) percentile, medium (90th) percentile, and high (90th) percentile, respectively. Another rise in peak percentage was observed in annual electricity generation where there was an upsurge of 6.80 and 6.24 for the 2050s medium (90th) percentile and high (90th) percentile, respectively, and 10.83, 13.85, and 17.45 for the 2080s low (90th) percentile, medium (90th) percentile, and high (90th) percentile, respectively. The analysis of cooling and heating energy for current weather and future projections identifies perhaps the most drastic effect of temperature on the overall consumption in the supermarket.

For cooling energy consumption in the supermarket, the peak percentage increase observed was 12.67 and 11.63 for the 2050s medium (90th) percentile and high (90th) percentile, respectively, and 20.15, 25.72, and 32.38 for the 2080s low (90th) percentile, medium (90th) percentile, and high (90th) percentile, respectively. For heating energy consumption in the supermarket, the peak percentage decrease observed was 47.37 for both the 2050s medium (90th) percentile and high (90th) percentile and 68.42, 73.68, and 84.21 for the 2080s low (90th) percentile, medium (90th) percentile, and high (90th) percentile, respectively.

Analysis of the external temperature, dry bulb temperature, and cooling load profile of the three warmest months of the year with clear sky for the current weather and the worst-case scenario data sets showed an increase of 19.22%, 6.33%, and 12.62%, respectively. However, in the case of heating load profile for the three coldest months of the year, there is a sharp reduction of 36.72%. It shows that the varying climatic weather projection affects the temperature of the supermarket directly including the cooling and heating profile, which is necessary to keep the supermarket temperature mode under the set temperature limit.

All these variations are in line with the range of annual average temperature change predicted by the general circulation model based on the IPCC scenarios, which generally shows an increase in temperature over time

The study, therefore, establishes the significant impact of the variability of climatic patterns on a supermarket's building performance, taking into consideration the future timelines that are also directly related to the life span of the building. It further upholds the premise that predicted that an increase in future temperatures results in an increase in energy use for cooling and emissions but conversely leads to the reduction in heating demand; likewise, an increase in cooling demand has environmental implications as it results in an increase in electricity consumption leading to higher carbon emissions related to the operational carbon emissions of the building.

This work has shown that the use of building performance simulations along with various scenarios of future climatic projections can contribute towards the mitigation of the environmental implications to the built environment. It could potentially help the architecture and civil engineering community to have an environment friendly approach towards the extreme changing temperature. This along with conscious decision making and a pre-emptive climate policy can help the buildings to be ready for the drastic future climate change.

This research contributes towards predictability of the implications of future projects and would in turn assist the decision makers to make sound, sustainable, environmentally friendly, and effective decisions. It also enables a drive towards the achievement of a more secure and sustainable future with a clearer understanding of the prerequisites required to build more futureproof supermarket buildings. It would give the decision makers opportunity to get ahead of the curve and adopt a more realistic plan to have the built environment secured against extreme weather change.

The focus on reducing the cooling loads in the future climate and improving the efficiency of the supermarket building will present a challenge to the innovators with most supermarkets leaning to adapt to the renewable and microgeneration technological advances. This technology would cater for the future increasing temperatures and adapt to the changing climate by acting as energy efficiency measures. It would also require better planning and design options to build a robust building design with proper equipment installation, high-efficiency HVAC systems, the introduction of passive design technologies to mitigate mechanical ventilation, and usage of better refrigerant with low environmental impact and excellent thermodynamic performance to reduce the future energy demands in the supermarkets.

**Author Contributions:** A.B.-J., A.M., and M.F. conceived and designed the project; A.H. performed the experiments and analysed the data. A.H. and A.B.-J. wrote the paper. A.B.-J., A.M., M.F., and H.T. reviewed the paper. All authors have read and agreed to the published version of the manuscript.

**Funding:** This research received no external funding.

**Conflicts of Interest:** The authors declare no conflict of interest.

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
