# Peer review of "Investigating the Potential Impact of Future Climate Change on UK Supermarket Building Performance"

_sustainability, doi:10.3390/su13010033_

Round 1
Reviewer 1 Report
Dear Authors,
The article is very interesting.
Please improve your drawings to make them more readable.
Reviewer 2 Report
The Article investigates the implications of future climatic conditions on the operation of 18 supermarkets in the UK. Below are some indications relating to the text submitted to the revision process.
Line 84 shows an open then not closed parenthesis
Line 85 the definition of TMY is approximate (it is always good practice for reference to the technical standard, in this case the reference document is EN ISO 15927-4).
Perhaps the premise would be completed by a reference to the policies undertaken by the EU with regard to European policies for reducing energy consumption.
Lines 138-147 It is not clear how the TMY files representing future climatic conditions were generated (Was only the temperature changed? How were the other climatic parameters treated?)
Line 170 Specify the type of mathematical model on which it is based the “TAS EDSL software version 9.5.0”. Is the model based on European technical standard? Or is it based on other models such as TRNSYS or EnergyPlus? More details would make the text clearer.
Line 214 Include information regarding the thermophysical characteristics of the opaque and transparent envelope elements (see ie EN ISO 13786 and characteristics of the transparent building envelope). Possibly even more details on the type of technical systems and how they were modelled.
Line 274 Associate the annual increases in energy consumption with an annual variation in climatic conditions (not only air-temperature, but also solar radiation and other climatic parameters). Furthermore, you should clearly indicate the reference cities to which the energy simulations refer. Also, perhaps, the results would be more readable in the form of a graph.
Line 295 It is not clear how the other climatic parameters vary
Line 284 The article refers to the increase in energy consumption for cooling and then this information is not examined among the results.
Line 328 Reference is made to tables 79 and 74 but they are not present in the text (perhaps there are no summary reports of the results).
Fig. 12, Fig. 13, Fig. 14 The figures are not very clear. Perhaps, to improve readability, it would make more sense to report statistical data (eg absolute frequencies by classes) and compare the different scenarios with each other.
Fig. 17-18 Figure unclear (make graphics more readable by using ie colours)
Conclusions If on the one hand there is an increase in energy consumption related to climatic variations, are there also variations connected to the production of renewable thermal and electrical energy due for example to the presence of more solar radiation (or less cloud cover)? There are no references regarding energy production from renewable energy. How could it vary over time?
Round 2
Reviewer 2 Report
The comments have been accepted and reported in the text, by inserting the changes the text is more complete and better presented. As a general rule, when carrying out research it is always good practice to refer to the technical regulations.
In Table 2. C it is better to refer to the units of measurement of the international system (i.e. in the Conductance use the Kelvin).
To describe the characteristics of the envelope, it would have been interesting to include indications on parameters provided for by the technical regulation (eg Internal areal heat capacity).
In table 5 it would have been better to use the same units of measurement when it comes to energy (kWh and MJ were used).
Match the font style used in images with that used in text. Some titles are overly large.